# Cross-Talk between Mitochondrial Dysfunction-Provoked Oxidative Stress and Aberrant Noncoding RNA Expression in the Pathogenesis and Pathophysiology of SLE

**DOI:** 10.3390/ijms20205183

**Published:** 2019-10-19

**Authors:** Chang-Youh Tsai, Song-Chou Hsieh, Cheng-Shiun Lu, Tsai-Hung Wu, Hsien-Tzung Liao, Cheng-Han Wu, Ko-Jen Li, Yu-Min Kuo, Hui-Ting Lee, Chieh-Yu Shen, Chia-Li Yu

**Affiliations:** 1Division of Allergy, Immunology & Rheumatology, Taipei Veterans General Hospital & National Yang-Ming University, #201 Sec.2, Shih-Pai Road, Taipei 11217, Taiwan; darryliao@yahoo.com.tw; 2Department of Internal Medicine, National Taiwan University Hospital, #7 Chung-Shan South Road, Taipei 10002, Taiwan; hsiehsc@ntu.edu.tw (S.-C.H.); b89401085@ntu.edu.tw (C.-S.L.); chenghanwu@ntu.edu.tw (C.-H.W.); dtmed170@yahoo.com.tw (K.-J.L.); 543goole@gmail.com (Y.-M.K.); tsichhl@gmail.com (C.-Y.S.); 3Institute of Clinical Medicine, National Taiwan University College of Medicine, #7 Chung-Shan South Road, Taipei 10002, Taiwan; 4Division of Nephrology, Taipei Veterans General Hospital & National Yang-Ming University, #201 Sec. 2, Shih-Pai Road, Taipei 11217, Taiwan; thwu@vghtpe.gov.tw; 5Section of Allergy, Immunology & Rheumatology, Mackay Memorial Hospital, #92 Sec. 2, Chung-Shan North Road, Taipei 10449, Taiwan; htlee1228@gmail.com

**Keywords:** noncoding RNA, microRNA, long noncoding RNA, mitochondrial dysfunction, oxidative stress, nitrosative stress. exosome, cross-talk, systemic lupus erythematosus

## Abstract

Systemic lupus erythematosus (SLE) is a prototype of systemic autoimmune disease involving almost every organ. Polygenic predisposition and complicated epigenetic regulations are the upstream factors to elicit its development. Mitochondrial dysfunction-provoked oxidative stress may also play a crucial role in it. Classical epigenetic regulations of gene expression may include DNA methylation/acetylation and histone modification. Recent investigations have revealed that intracellular and extracellular (exosomal) noncoding RNAs (ncRNAs), including microRNAs (miRs), and long noncoding RNAs (lncRNAs), are the key molecules for post-transcriptional regulation of messenger (m)RNA expression. Oxidative and nitrosative stresses originating from mitochondrial dysfunctions could become the pathological biosignatures for increased cell apoptosis/necrosis, nonhyperglycemic metabolic syndrome, multiple neoantigen formation, and immune dysregulation in patients with SLE. Recently, many authors noted that the cross-talk between oxidative stress and ncRNAs can trigger and perpetuate autoimmune reactions in patients with SLE. Intracellular interactions between miR and lncRNAs as well as extracellular exosomal ncRNA communication to and fro between remote cells/tissues via plasma or other body fluids also occur in the body. The urinary exosomal ncRNAs can now represent biosignatures for lupus nephritis. Herein, we’ll briefly review and discuss the cross-talk between excessive oxidative/nitrosative stress induced by mitochondrial dysfunction in tissues/cells and ncRNAs, as well as the prospect of antioxidant therapy in patients with SLE.

## 1. Introduction

Systemic lupus erythematosus (SLE) is a highly heterogeneous disorder with chronic inflammatory and autoimmune reactions all over the body. It is characterized by the production of diverse autoantibodies [1,2] and chronic tissue inflammation [3,4,5,6]. There are multiple factors associated with lupus pathogenesis, including genetic predisposition [7,8,9,10,11,12,13,14,15], epigenetic dysregulation of gene transcription [16,17,18,19,20,21] and aberrant post-transcriptional events by noncoding (nc)RNAs [19,22,23,24,25], sex hormonal imbalance [26,27,28,29], environmental stimulation [30,31], mental/psychological stresses [28], dietary/nutritional influence [32,33,34,35], mitochondrial dysfunctions [36,37,38,39], and other yet-undefined factors [40]. Figure 1 shows the factors contributing to the pathogenesis of SLE, in which environmental factors such as infections, chemicals, heavy metals, medications, exogenous estrogens, and phthalate trigger its development in susceptible individuals. The genome-wide association study (GWAS) has identified over 100 risk loci for SLE susceptibility across populations [13]. However, functional studies have revealed that many of them fall in the category of noncoding regions of genomes, suggesting that they probably play a regulatory role. Many loci exhibit protean environmental interactions, epigenetic modifications, or association with genetic variants [10]. Nevertheless, the expression of IFN-α in tissues and circulation has been consistently found at a hereditary risk locus in patients with SLE [14]. The genetic predisposition for lupus pathogenesis is summarized in Table 1.

Recent investigations revealed that increased oxidative and/or nitrosative stress could induce structural and functional changes in different biomolecules, including proteins, lipids, nucleic acids, and glycoproteins [41,42]. The oxidative stress may also modulate proinflammatory cytokine gene expression [43,44,45,46] and cell senescence/apoptosis [47,48]. Antioxidants have been tried in the treatment of SLE with effectiveness [49,50,51,52,53]. Accordingly, the presence of oxidative stresses and their associated biomarkers are definitely playing a decisive role in the pathogenesis of SLE [54].

Epigenetics is an investigation of the changes in phenotypic presentation (or gene expression) that are caused by mechanisms other than the polymorphism of genome per se. It is conceivable that more than 97% of cellular RNAs are not transcribed for protein coding in nature. These ncRNAs, including microRNAs (miRs, 20–24 bp in length) and long noncoding (lnc) RNAs, which are >200 bp in length are the major molecules for post-transcriptional modifications of messenger (m)RNAs [55,56]. Interestingly, many reports have demonstrated that oxidative stress can modulate ncRNA expression in different diseases [57,58]. Conversely, ncRNAs have also been found to be regulators of oxidative stresses in different pathological conditions [59]. Furthermore, the cross-talk between miRs and lncRNAs has also been found [60,61]. Based on these facts, we hereby review and discuss briefly the molecular basis of epigenetic regulations, the underlying mechanism of mitochondrial dysfunctions, and the cross-talk between mitochondrial dysfunction-provoked oxidative stress and abnormal expression of ncRNAs during the pathologic development of SLE. At the end, a potential use of antioxidants as the therapy for SLE will also be concisely overviewed.

## 2. Epigenetic Regulations of Gene Expression/Silencing in Physiological Conditions

Epigenetic variation is a reversible but heritable change in gene expression without alterations in genetic code. It may include DNA methylation, histone modification, and post-transcriptional mRNA modification by ncRNAs [16]. DNA methylation is a biochemical process that involves a methyl group being added to a cytosine or adenine residue at the position of a repeated CpG dinucleotide (CpG island) in the promoter region to repress gene expression by DNA methyl- transferase (DNMT) 1, 3a, and 3b. In contrast, reactivation of DNA by demethylation to restore gene transcription can be achieved by ten-eleven translocation (TET) enzymes TET1, TET2, and TET3.

### 2.1. Abnormal DNA Methylation/Demethylation in SLE

DNA methylation is catalyzed by DNMT1 for gene silencing. A status of DNA hypomethylation to enhance gene expression can be found in CD4^+^T cells of SLE patients as a result of decreased expression of DNMT1 originating from a deficient *ras*-*MAPK* signature [62,63]. In addition, DNA methylation acts as a housekeeping mechanism for physiological inactivation of X-chromosomes in female [26,27,64]. Recent studies have suggested that C*D40L* demethylation is responsible for CD40L overexpression in T cells of women with SLE [64].

### 2.2. Abnormal Histone Modification in SLE

The degree of chromatin tightness is regulated via complex mechanisms, including structural changes in histones. Usually, double helix-chromatin coils around a protein core composed of histone octamers (H2A, H2B, H3, and H4 with two copies of each). The biochemical processes to change the 3D structure of histones include ubiquitination, phosphorylation, SUMOylation, methylation, and acetylation. The methylation and acetylation of histones are the most extensively studied [17]. These two biochemical changes are controlled by two major enzymes, histone acetyl transferase (HATs) and histone deacetylase (HDACs), that catalyze the addition/removal of an acetyl group on the lysine residues of histones. Acetylation relaxes the chromatin structures by diminishing the electric charge between histone and DNA as a result of offering an acetyl group. Conversely, deacetylation tightens the chromatin structure to silence gene expression.

The participation of histone modifications in lupus pathogenesis has been well documented. Hu et al. [65] demonstrated a global hyperacetylation of histones H3 and H4 in lupus CD4^+^T cells. Zhou et al. [66] reported that abnormal histone modifications within TNFSF7 promotor caused CD70 (a ligand for CD27) overexpression in SLE-T cells. Furthermore, Hedrich et al. [67] demonstrated that CREM, a transcription factor, participated in histone deacetylation in active T cells of SLE patients by way of silencing IL-2 expression, which normally recruits HDAC to cis-regulatory element (Cre) sites in IL-2 promotors. Dai et al. [68] showed in GWAS an alteration in histone H3 lysine K4 trimethylation (H3K4me3) by chromatin immunoprecipitation linked to microarray in peripheral blood mononuclear cells of some SLE patients. In addition, Zhang et al. [69] have found global H4 acetylation occurs in monocytes/macrophages in SLE subjects, which is regulated by IFN regulatory factors. The release of SLE-related cytokines such as IL-17, IL-10, and TNF-α was also abnormally increased in H3 acetylation by *stat*3 [70,71,72]. In lupus-prone MRL/*lpr* mice, a histone deacetylation gene, *sirtuin-1* (*Sirt*-1), was found overexpressed [73], indicating a compensatory repression of gene over-reactivation. Hu et al. [73] further noted downregulation of *Sirt*-1 would transiently enhance H3 and H4 acetylation and subsequently mitigate serum levels of anti-dsDNA, as well as kidney damage in lupus mice. Javierre et al. [74] reported a global decrease in the 5-methylcytosine content in parallel with DNA hypomethylation and high expression levels of ribosomal RNA genes relevant to SLE pathogenesis. In short, abnormal histone modifications are implicated in lupus pathogenesis and immunopathological changes in these patients.

### 2.3. Physiological Functions of ncRNAs

Besides DNA methylation/acetylation and histone modification, the most recently discovered epigenetic mechanisms for gene expression are dependent on the class of ncRNAs that are not translated into proteins. These molecules include both housekeeping ncRNAs and regulatory ncRNA [55]. In total 50% of mRNAs are located in chromosomal regions with liability to undergo structural changes [75]. On the other hand, lncRNA can regulate gene expression by different ways, including epigenetic, transcriptional, post-transcriptional, translational, and peptide localization modifications [56]. Interestingly, the interactions between lncRNAs and miRs, as well as their pathophysiological significance, have recently been reported [60,61]. It is believed that lncRNAs mediate “sponge-like” effects on various miRs and subsequently inhibit miR-mediated functions [60,61]. The regulatory effects of intracellular and extracellular (exosomal) ncRNA on cell functions are illustrated in Figure 2.

### 2.4. Aberrant Intracellular and Extracellular Exosomal ncRNA Expression in Association with Pathological Changes in Patients with SLE

It is not surprising that miRs play important roles in the regulation of innate and adaptive immunity, and the aberrantly expressed miRs are associated with autoimmune diseases [22,76,77,78,79,80]. Lu et al. [23,81,82,83] and Su et al. [84] have found various aberrantly expressed intracellular miRs implicated in the cell signaling abnormalities, deranged cytokine and chemokine release, and Th17/Treg ratio alterations in patients with SLE. Different from miRs, lncRNAs are expressed at lower levels in cells and tissues, more specifically [85,86,87]. These lncRNA are obviously modulating innate immunity [88] and inflammatory responses [89]. Luo et al. [90], Zhao et al. [91], and Wang et al. [92] reviewed the literature and found that lncRNA expression profiles in SLE were remarkably different from the normal.

The regulatory functions of miRNAs can be validated by transfecting miRNA mimics or antagonists using electroporator. Lu et al. [81] found increased miR-224 could target apoptosis inhibitory protein 5 (API5) and enhance T cell activation, and then activate induced cell apoptosis. Besides, the same group found decreased miR-31 in SLE T cells targeted the *Ras* homologue gene family member A (*Rho*A), which led to a decreased nuclear factor of activated T cells (NFAT) and cell apoptosis [23]. In addition, decreased miR-146a may result in upregulation of interferon regulatory factor 5 (IRF-5) and then enhanced production of IFN-α, STAT-1, IL-1 receptor associated kinase-1 (IRAK1), and TRAF6, which then increase innate immune responses, lupus disease activity, and lupus nephritis [23]. Furthermore, increased miR-524-5p that targets Jagged-1 and Hes-1mRNA may enhance IFN-γ production and then increase disease activity of SLE [82]. Su et al. [84] demonstrated that increased expression of miR-199-3p promoted ERK-mediated IL-10 production by targeting poly-(ADP-ribose) polymerase-1 (PARP-1) in SLE.

While their major functions are executed intracellularly, many miRs can be detected extracellularly in plasma/serum and urine. This extracellular form of ncRNA is protected from degradation by conjugation with carrier proteins or by being enclosed in subcellular vesicles by lipid bilayer exosomes [85]. With characteristics of the tissue- and disease-specific expression, these extracellular ncRNAs can carry out intercellular communication, signal transduction, transport of genetic information, immunomodulation, and can be taken as diagnostic biosignatures or as research tools for understanding the pathophysiology of autoimmune diseases [85,86,87,88,89,90,91,92]. Plasma circulating microRNAs exist in a rather stable form and are incorporated into distant cells to regulate protein translation and synthesis there. Carlsen et al. [87] have found plasma exosomal miR-142-3p, which targets IL-1β, and miR-181a, which targets FoxO1, are increased in active SLE patients. Kim et al. [88] demonstrated that increased plasma circulatory hsa-miR-30e-5p, hsa-miR-92a-3p, and hsa-miR-223-3p could become novel biosignatures in patients with SLE. The exosomal miRs can be found in other body fluids including breast milk, saliva, and urine, in addition to plasma [89]. Hsieh et al. [93] and Tsai et al. [94] concluded that urinary exosomal miRs could be used as biomarkers/biosignatures in lupus nephritis. Tsai et al. [94] have also noted aberrant miRNA expression in the immune-related cells could become biosignatures in correlation with pathological processes in different autoimmune and inflammatory rheumatic diseases. In addition, Perez-Hernandez et al. [95] and Xu et al. [96] have suggested the potential therapeutic application of exosomal ncRNA in different autoimmune diseases. Not only exosomal miRs, extracellularly expressed lncRNA profiles could also become potential biomarkers for human diseases [97,98]. lncRNAs are another regulatory noncoding RNA, capable of modulating many biological functions more specifically than miRs [99,100,101,102]. Aberrant expression of lncRNAs obviously induces different disease entities [99,100,101,102,103,104,105,106]. Table 2 summarizes the aberrant intracellular and circulating plasma exosomal lncRNA expression, their target mRNA, and related pathological processes in patients with SLE. Wang et al. [103] found that increased lncRNA ENST00000604411.1 expression in macrophages/dendritic cells, through targeting the X inactive specific transcript (XIST) that is normally implicated in keeping the active X chromosome in an activated state by protecting it from ectopic silencing after commencement of the silencing process of the haplotype X chromosome, could induce lupus development. Another lncRNA ENST 00000501122.2 (also known as NEAT1) overexpressed in SLE monocytes may activate CXCL-10 and IL-6 expression. Furthermore, Wu et al. [98] reported that elevated expression of plasma GAS-5, linc 0640, and linc 5150 may activate MAPK signaling pathway. The five lncRNA panels, including GAS-5, linc7074, linc 0597, linc 0640, and linc 5150 in plasma, could be regarded as biosignatures in SLE. The biochemical properties of extracellular ncRNAs and the pathophysiological roles of these aberrant exosomal ncRNAs in SLE are further discussed in the following paragraph.

## 3. Increased Oxidative Stress in Patients with SLE

### 3.1. Causes of Excessive Oxidative Stress in SLE

Li et al. [108] have compared the reduction–oxidation (redox) capacity between normal and SLE immune cells. They found decreased plasma and intracellular glutathione (GSH) levels, and decreased intracellular GSH-peroxidase and gamma-glutamyl-transpeptidase activity in patients with SLE. Besides, the defective expression of facilitative glucose transporter (GLUT) 3 and 6 led to increased intracellular basal lactate levels, as well as decreased ATP production in SLE T cells and polymorphonuclear leukocytes. These results may indicate deranged cellular bioenergetics and defective redox capacity in immune cells that would increase oxidative stress in SLE. Lee et al. [36,37,38,39] demonstrated that mitochondrial dysfunctions in SLE patients included decreased mitochondrial DNA (mtDNA) copy number, increased mtDNA D-310 (4977 bp) heteroplasmy, and variants, as well as polymorphism of C_1245_G in *hOGG*1 gene in leukocytes. Leishangthem et al. [41] found a significant decrease in enzyme activity of complex I, IV, and V in mitochondria of patients with SLE. Lee et al. [109] have extensively investigated the cause of excessive stress in patients with SLE. They reported a number of antioxidant enzyme deficiencies in SLE leukocytes, including copper/zinc superoxide dismutase (Cu/ZnSOD), catalase, glutathione peroxidase 4 (GPx-4), glutathione reductase (GR), and glutathione synthetase (GS). In addition, the mitochondrial biogenesis-related proteins, such as mtDNA-encoded ND1 peptide (ND1), ND6, nuclear respiratory factor 1(NRF-1), and pyruvate dehydrogenase E1 component alpha subunit (PDHA1), and glycolytic enzymes, including hexokinase II (HK-II), glucose 6-phosphatate isomerase (GPI), phosphofructokinase (PFK), and glyceraldehyde 3-phosphate dehydrogenase (GAPDH), are also reduced in SLE immune cells. These mitochondrial functional abnormalities may further increase oxidative stress and cell apoptosis in patients with SLE, in addition to the defective bioenergetics. Yang et al. [110] and Tsai et al. [111] concluded that enhanced oxidative stress could facilitate mitophagy, inflammatory reactions, cell senescence/apoptosis, neoantigen formation, and NETosis in SLE. The causes of mitochondrial dysfunction to induce excessive oxidative stresses and their effects on the lupus pathogenesis and pathological processes are illustrated in Figure 3.

### 3.2. Effects of Excessive Oxidative Stress on the Pathogenesis and Pathophysiology in SLE Patients

The modifications of intra- and extracellular biomolecules by oxidative stress result in glycation and nitrosation of proteins [112], lipid peroxidation [42], as well as mitochondrial [113] and nuclear DNA strand breaks [114]. These biochemical and structural modifications of intracellular biomolecules would induce histone modification, nuclear and mitochondrial DNA damage, and aberrant ncRNA expression. As a consequence, the resulting sensitivity to environmental stress and sex hormone dysregulation [26,27,28,29,30,31] may further trigger the occurrence of lupus flare-ups. In addition, cardiovascular morbidities are enhanced due to increased glycation end products in patients with SLE [111,112,115]. The molecular basis and adverse effects of excessive oxidative stress in lupus pathogenesis and pathology are summarized in Figure 4.

## 4. Cross-Talk Between Oxidative Stress and ncRNAs in Physiological Condition

Recently, ever-increasing studies have emphasized the significance of the interactions between redox signaling and expression of ncRNAs in normal physiological conditions, as well as in disease status [44,45,46,57,58,59]. Sustained high levels of oxidative stress can cause cell senescence and even cell death, while optimal oxygen radicals are important for cell signaling. Dandekar et al. [44] and Lin et al. [116] have found mutual cross-talk among endoplasmic reticulum stress, oxidative stress, inflammatory response, and autophagy.

### 4.1. Excessive Oxidative Stress May Influence ncRNA Expression in Various Diseases

Many authors have demonstrated that redox-dependent signaling is essential for host’s cellular decisions on differentiation, senescence, or death to maintain homeostasis of the body [117,118,119]. Figure 5 summarizes the aberrant miR expression resulting from excessive oxidative stress in different diseases, which include Alzheimer’s disease [120], Parkinson’s disease [121], hearing disorders [122], aging [123], osteoarthritis [124], cardiomyopathy in diabetes [125], and cancers [126]. However, despite the association of aberrant ncRNA expression with various pathological changes in SLE, as listed in Table 2 and Table 3, there has been no literature demonstrating direct evidence for specific oxidative-induced ncRNA in patients with SLE. The combination of Table 3 and Figure 5 leads us to speculate that miR-21, miR-29b, miR-146a, and miR-126b may be induced by excessive oxidative stress in SLE as asterisked in Table 3 and its footnote.

### 4.2. Aberrant ncRNA Expression Induces Oxidant/Antioxidant Imbalance in Different Pathological Processes

It has been demonstrated that excessive oxidative stress can affect ncRNA expression in Section 4.1. However, it is quite interesting that aberrant expression of ncRNAs conversely regulates redox balance in some pathological conditions. Esposti et al. [127] found miR-500a-5p could modulate oxidative stress-responsive genes in breast cancer and predict breast cancer progression as well as survival. Sangokoya et al. [128] have demonstrated that miR-144 modulates oxidative stress tolerance and, thus, is associated with changes in anemia severity in sickle cell disease. Kim et al. [129] found the roles of lncRNA and RNA-binding proteins in oxidative stress, cellular senescence, and age-related diseases. Tehrani et al. [130] further demonstrated multiple functions of lncRNAs in regulating oxidative stress, DNA damage response, and cancer progression. Mechanistically, ncRNAs can regulate enzymatic activity of different glutathione S-transferases (GSTs) to affect redox homeostasis [58]. These GSTs include microsomal GST, GST zeta l, GST mu1, GST theca 1, and sirtuin 1, superoxide dismutase 2 and thioredoxin reductase 2. In addition, the cellular oxidant/antioxidant balance can also be regulated by lncRNAs [59]. The abnormal ncRNA expression to affect the oxidant/antioxidant system is summarized in Figure 6.

## 5. Antioxidant Therapy and Manipulation of Epigenetic Expression to Treat Patients with SLE

In addition to increased oxygen free radicals in the plasma of SLE patients, there are other novel findings regarding the pro-oxidant/antioxidant balance in SLE. Mohan et al. [131] firstly confirmed that plasma concentrations of lipid peroxidase and nitric oxide were increased, whereas antioxidant molecules such as catalase, superoxide dismutase (SOD), GSH peroxidase, and vitamin E were decreased. Obviously, the pro-oxidant/antioxidant balance in SLE is disturbed [53]. Antioxidant therapy has been advocated for ameliorating tissue damage caused by excessive pro-oxidant radicals. Supplemented with GSH precursor, N-acetyl-cysteine (NAC) can improve disease activity in lupus-prone mice [50]. Delivering the *oxidation resistance*-1 (*OXR*1) gene to mouse kidneys by genetic manipulation can protect the kidney from damage induced by serum nephrotoxic agents, and prevent the animal from developing lupus nephritis [52]. Many authors, by administering NAC, have found remedies to ameliorate lupus activities in human SLE. Kudaravalli et al. [132] reported the improvement of endothelial dysfunction in patients with SLE by NAC and atorvastatin. Lai et al. [133] reported that NAC reduced disease activity by blocking mammalian targets of rapamycin (mTOR) in T cells of SLE patients. Tzang et al. [134] found cystamine attenuated lupus-associated apoptosis in ventricular tissue by suppressing both intrinsic and extrinsic apoptotic pathways. Nevertheless, much more clinical data are necessary to validate the efficacy of antioxidant therapy in managing patients with SLE.

Since there are so many intricate interactions among oxidative/nitrosative stress, epigenetic regulations, and gene expression in SLE, as discussed in the above sections, interference with epigenetic mechanisms such as modifying the activity of histone acetylase and/or DNA methylation, or inducing up- or downregulation of ncRNA expression may be helpful and can also be advocated to detour lupus pathogenesis and to diminish SLE disease activity in the future [135,136].

## 6. Conclusions

Mitochondrial dysfunction-provoked excessive oxidative stress is a crucial downstream contributory factor for lupus pathogenesis in addition to the dysregulation of upstream genetic/epigenetic functions. Recent studies have revealed that mutual interactions between oxidative stress and epigenetic regulation can perpetuate pathogenesis and pathological processes in SLE and other autoimmune diseases, as well as ageing-related diseases. In the ncRNA regulatory system, cross-talk between lncRNAs and miRs can occur for fine tuning of gene expression. Excessive oxidative stress-derived ROS and RNS may trigger autoimmune reaction and increase cell senescence/cell death in lupus-susceptible individuals. Antioxidant therapy and epigenetic modulators might become novel therapeutic strategies to treat SLE in the future.

## Figures and Tables

**Figure 1 ijms-20-05183-f001:**
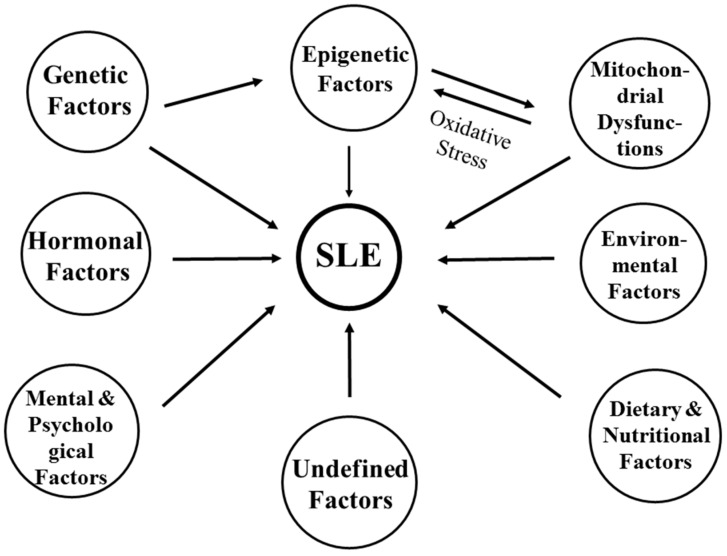
Factors contributing to the development of systemic lupus erythematosus. It is worthy to note that cross-talk between mitochondrial dysfunction and aberrant epigenetic regulation is mediated via excessive oxidative stress.

**Figure 2 ijms-20-05183-f002:**
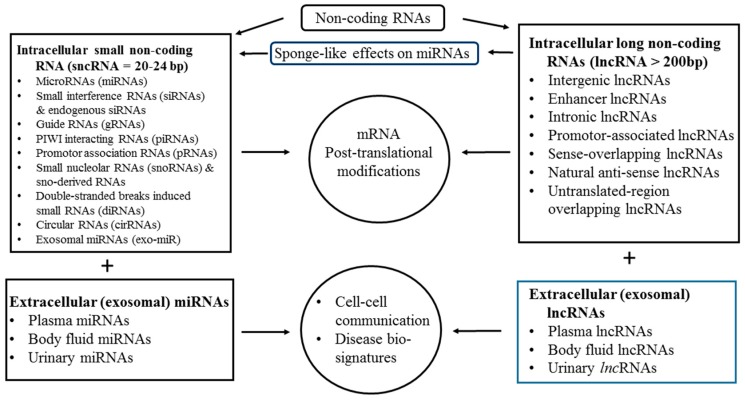
Different kinds of noncoding RNAs, including groups of small noncoding and long noncoding RNA, distributed in the intracellular and extracellular compartments, such as plasma, urine, and other body fluids, for regulation of messenger RNA translation and remote cell–cell communications in the body.

**Figure 3 ijms-20-05183-f003:**
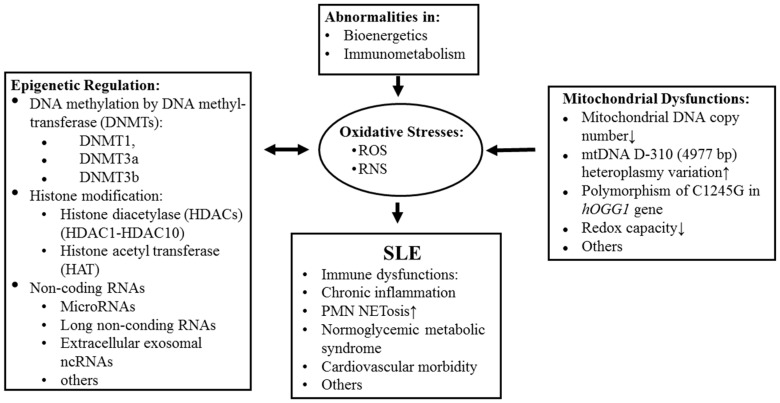
The origins of excessive oxidative stresses and their roles in abnormal epigenetic regulation and pathological processes in patients with SLE.

**Figure 4 ijms-20-05183-f004:**
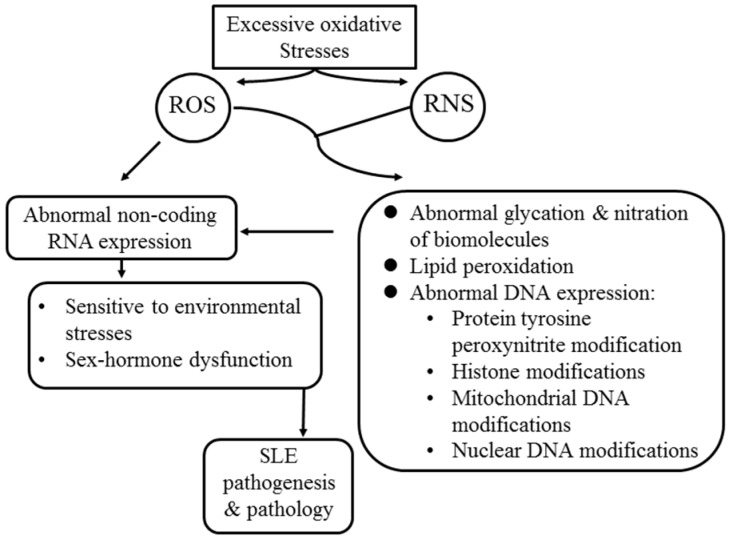
The molecular basis of excessive oxidative stress in the pathogenesis and pathological changes in patients with SLE.

**Figure 5 ijms-20-05183-f005:**
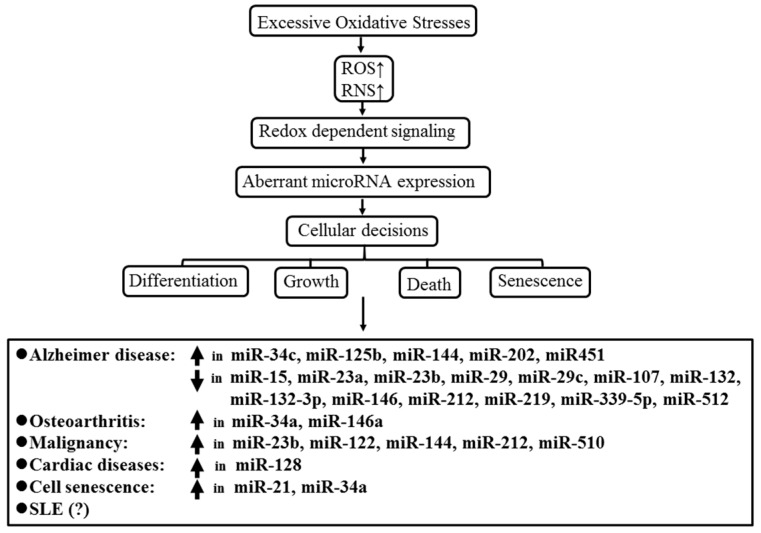
The effect of excessive oxidative stress on aberrant microRNA expression in various degenerative, malignant, cardiovascular, and autoimmune diseases. (?): increased miR-21, miR-29, miR-126b, and miR-146a expression induced by excessive oxidative stress is suspected in SLE patients, but no direct evidence has been published in the literature.

**Figure 6 ijms-20-05183-f006:**
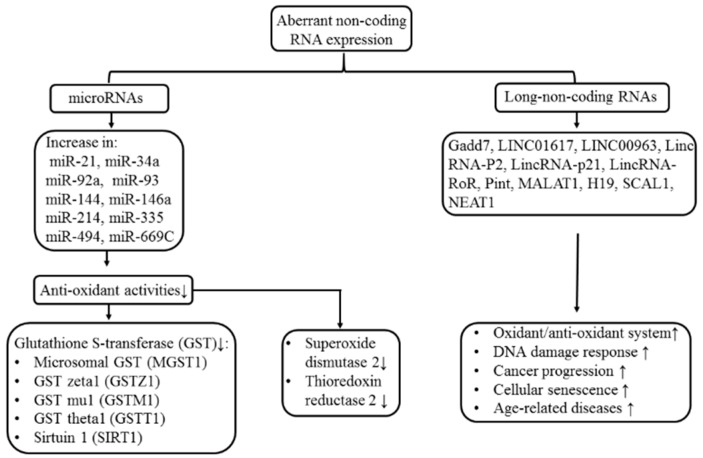
The effects of aberrant noncoding RNA expression on redox capacity and the induction of various age-related and malignant diseases.

**Table 1 ijms-20-05183-t001:** Some of the genetic loci involved in the risk for SLE.

**MHC association** [7,8,9] - MHC class II: DR_2_, DR_3_ - MHC class III: C_4_ null, TNF-α **Immune complex processing and phagocytosis** [7,8,9,10,11,12,13,14,15] - C1_q/r/s_, C_4A/B,_ CFB - *FCGR*2A/B, CR2, CR3 - CRP - *ICHM*s (intercellular adhesion molecules) - *ITGAM* (integrin subunit alpha M) **TLR and type I IFN signaling** [7,8,9,10,11,12,13,14,15]: - *TLR*7 (toll-like receptor 7) - *TREX*1 (three prime repair exonuclease 1) - *DNASE*1 (DNA degrading enzyme 1) - *IRAK*1/*MECP*2 (interleukin-receptor-associated kinase 1) - *IRF*5/7/8 (interferon regulatory factor 5, 7, 8) - *STAT*1 (signal transducer and activator of transcription 1) - *STAT*4 (signal transducer and activator of transcription 4) **B and T cell function and signal genes** [7,8,9,10,11,12,13,14,15] - *IL*10 (interleukin 10) - *STAT*4 (signal transducer and activator of transcription 4) - *PTPN*22 (protein tyrosine phosphatase non-receptor type 22) - *PDCD*1 (programmed cell death 1) - *TNFSF*4 (TNF superfamily member 4) - *BLK* (B lymphoid tyrosine kinase) - *BANK*1 (B cell scaffold protein with ankyrin repeats 1) **Others** - *PXK*/*ABHD*6 (PX domain containing serine/threonine kinase likes) - *XKR*6 (XK related 6) - *UPF*1/*SMG*7 (RNA helicase and ATPase) - *NMNAT*2 (nicotinamide nucleotide adenyltransferase 2) - *UHRF*1*BP*1 (ubiquitin like with PHD and ring finger domains 1 binding protein 1)

**Table 2 ijms-20-05183-t002:** Aberrant expression of long none-coding RNAs, their target mRNAs, and related pathological processes in patients with systemic lupus erythematosus.

SLE	lnc RNA Expression	Target mRNA	Pathological Processes
**Intracellular** [103,104,105,106]			
	NEAT_1_↑*	IL-6↑, IFN↑, CXCL10↑	DNA hypomethylation
	MALAT_1_↑	IL-21↑, SIRT_1_↑	SLEDAI-2K↑
	Linc0597↑	TNF-α↑, IL-6↑	ESR↑, CRP↑, C3 ↓,
	Linc DC↑	STAT3↑	Th1↑
	ENST00000604411.1↑	XIST	SLEDAI score↑
	ENST000005011222↑	NEAT_1_	
	Linc 0949↓	TNF-α↑, IL-6↑	Inflammation↑
	Linc-HSFY2-3:3↓	-	SLEDAI score↑
	Linc-SERPIN139-1:2↓	-	
	Gas 5↓	Apoptotic gene↓	T cell apoptosis↓
**Circulating plasma exosomal** [98]			
	Linc0597↑	TNF-α↑, IL-6↑	MAPK signaling↑
	Lnc0640↑	Phosphatase 4 (DUSP4)↑	Lupus pathogenesis
	Lnc5150↑	Arrestin β2 (ARRB_2_)↑	
		Ribosomal protein S_6_ kinase A_5_ (RPS6KA5)↑	
	Gas 5↓	Apoptotic gene↓	T cell apoptosis↓
	Lnc 7074↓		

↑: increased expression or production; ↓: decreased expression or production; *: Oxidative stress-induced [107].

**Table 3 ijms-20-05183-t003:** Aberrant expression of microRNAs, their target mRNAs, and pathological effects in patients with SLE.

SLE	miRNA		Target mRNA	Pathological Process
**Intracellular [82,83,84,85,86]**	● **Increase in:**			
		miR-21*	Arylamide small nucleotide inhibiors	DNA hypomethylation↑
		miR-524-5p	Jagged-1, Hes-1	IFN-γ↑, SLEDAI↑
		miR-126	KRAS	
		miR-148a	PTEN	
	● **Decrease in:**			
		miR-142-3p	HMGB-1	T and B activation↑
		miR-142-5p	PD-L1	
		miR-146a*	IRF-5, STAF-1	Innate immune response↑, lupus nephritis↑
		miR-224↑	API5	Type 1, IFN↑
		miR199-3p↑	PARP-1	IL-10↑
	● **Decrease in:**			
		miR-31	RhoA	Cell apoptosis↑
		miR-142-3p	HMGB-1	
		miR410	STAT3	
		miR-125a	STAT3, hexokinase 2, NEDDG	IL-10↑
		miR-125b*	Claudin 2, cingulin, SYVN1	
		mi-1273e		Th17/Treg ratio↑
		miR-3201		
**Circulating plasma [87,88,89,90,91,92,93,94]**	● **Increase in:**			
		miR-142-3p	IL-1β	
		miR-181a	FoxO1	
		hsa-miR-30e-5p hsa-miR-92a-3p		Oral ulcer and lupus anticoagulant
		hsa-miR-223-3p		
		miR-16-5p	*p*38MAPK, NF-κB	
		miR-223-3p	Voltage-gated K^+^ channel K_V4.2_	
		miR-451	LKB1/AMPK	
	● **Decrease in:**			
		miR-106a	THBS_2_	
		miR-17	JAB1/CSN5	
		miR-20a	IkBβ	
		miR-203	ZEB1	
		miR-92a	*p*63	
		miR-146a	JAK2/STAT3	
		miR-1202	cyclin dependent kinase 14	
**Urinary exosomal (lupus Nephritis) [95,96]**	● **Increase in:**			
		miR-125a	STAT3, hexokinase 2, NEDDG	Glomerulonephritis
		miR-146*	NF-κB	
		miR-150	Akt3	
		miR-155	PTEN, Wnt/β-catenin	
	● **Decrease in:**			
		miR-141	Tram1, GL/2, TGF-β	Glomerulonephritis
		miR-192	nin one binding protein	
		miR-200a	HMGB1/RAGE	
		miR-200c	ZEB1, Notch 1	
		miR-221	BIM-Bax/Bak, TIMP3	
		miR-222	PPP2R2A/Akt/mTOR, PCSK9	
		miR-429	TRAF6, DLC-1, HIF-1α	
	● **Decrease in:**			
		miR-3201		Endocapillary glomerular inflammation
		miR-1273e		

↑: increased expression or function; *: oxidative stress-induced microRNAs.

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
