# Peer review of "Cross-Talk between Mitochondrial Dysfunction-Provoked Oxidative Stress and Aberrant Noncoding RNA Expression in the Pathogenesis and Pathophysiology of SLE"

_ijms, 2019, doi:10.3390/ijms20205183_

Round 1
Reviewer 1 Report
The manuscript concerns a review that sheds light on the intricate relation of mitochondrial dysfunction, oxidative stress, ncRNA and SLE. There are some issues that need to be addressed before this manuscript can be accepted for publication.
Major comments:
In the manuscript, the authors should describe some examples about which ncRNA/downstream target gene (protein coding gene or miRNA)/downstream pathway were involved in pathogenesis of SLE (not just for this part). Meanwhile, these descriptions could match to the table. Table 1 and 2 should be re-organized. It would be easy to understand. What does it mean “?” in Table 1? What does it mean “Others” in Table 1? What does it mean “↑or ↓” in Table 2? These should be described or defined. The authors should add “miRNA target gene, oxidative stress and epigenetic regulation” to the header of this column. I suggested the descriptions of miRNA and lncRNA in table 2 could be separated. Were lnc0597, lnc0949, lnc7074, lnc5150, lnc0597 and lnc0640 the new lncRNAs? I have never seen these descriptions. The references in Table 2 should be re-checked. For example, information of lnc7074, lnc5150, lnc0597 and lnc0640 were not described in reference 103. The descriptions of section 2.3 and introduction (page 3, lines 96-109) were similar. Again, the name of ncRNA involved in exosome-mediated function should be described in detail. Which molecular/stimulator caused oxidative stress (OS) and which ncRNA was involving in OS-mediated function should be listed. Figure 5 should be described in detail. Which lncRNA was involved in epigenetic regulation should be described and listed in Table 2.
Minor comments:
IFN- should be re-checked. There are several typing errors throughout the manuscript, which should be carefully reviewed by authors before submission. “miR-200C” should be “miR-200c” in Table 2. “miR-34C” should be “miR-34c” in Figure 5.
Author Response
Answers to the Reviewer (1):
Major comments:
Question (1): In the manuscript, the authors should describe some examples about which ncRNA/downstream target gene (protein coding gene or miRNA)/downstream pathway were involved in pathogenesis of SLE (not just for this part). Meanwhile, these descriptions could match to the table.
Answer: Thanks for these important suggestions. We have already added 3 paragraphs to show the targets of the aberrant ncRNAs in SLE from the findings of ours and other authors’ publications in the Section 2.4, Table 2 and Table 3:
“Lu et al. [82] found increased miR-224 could target apoptosis inhibitory protein 5 (API5) and enhanced T cell activation and then activation induced cell apoptosis. Besides, the same group found decreased miR-31 in SLE T cells targeted Ras homologue gene family member A (RhoA) that lead to decreased nuclear factor of activated T cells (NFAT) and cell apoptosis [84]. In addition, Decreased miR-146a that targets interferon regulatory factor 5 (IRF-5) and then IFN-alpha, STAT-1, IL-1 receptor associated kinase-1 (IRAK1) and TRAF6 production those may enhance innate immune responses, lupus disease activity and lupus nephritis [84]. Furthermore, increased miR-524-5p that targets Jagged-1 and Hes-1mRNA may enhance IFN-gamma production and then increased disease activity of SLE [83]. Su et al [86] demonstrated that increased expression of miR-199-3p promoted ERK-mediated IL-10 production by targeting poly (ADP-ribose) polymerase-1 (PARP-1) in SLE.
“Carlsen et al. [89] found plasma exosomal miR-142-3p could target IL-1beta and miR-181a targeted FOXO1 for modulating innate immunity. Kim et al. [90] demonstrated that increased plasma circulatory hsa-miR-30e-5p, hsa-miR-92a-3p and hsa-miR-223-3p could become novel bisignatures in patients with SLE. The exosomal miRs can be found in other body fluids including breast milk, saliva and urinein addition to plasma [91].”
“Table 3 summarizes the aberrant intracellular and circulating plasma exosomal lncRNA expression, their target mRNA and related pathological processes in patients with SLE. Wang et al. [105] found that increased lncRNA ENST00000604411.1 expression in macrophages/ dendritic cells that targets X inactive specific transcript (XIST) might involve X chromosome inactivation through protecting the active –X from ectopic silencing and induces lupus pathogenesis. Another lncRNA ENST 00000501122.2 (also known as NEAT1) overexpressed in in SLE-monocytes may activate IL-6, CXCL-10 and IL-6 expression. Furthermore, Wu et al. [100] reported that elevated expression of plasma GAS-5, linc 0640 and linc 5150 may activate MAPK signaling pathway. The five lncRNAs panel including GAS-5, linc7074, linc 0597, linc 0640 and linc 5150 in plasma could be applied as biosignatures in SLE.”
Question 2: Table 1 and 2 should be re-organized. It would be easy to understand.
Answer: We appreciate for this valuable suggestion. We have already re-organized the Table 1. The new Table 1 looks clear and easy understanding. The new Table 2 has been re-arranged and divided into two Tables, new Table 2 and new Table 3. The new Table 2 summarizes the intracellular, plasma exosomal, and urinary exosomal microRNAs in SLE. The new Table 3 specifically deals with intracellular and plasma exosomal lncRNAs. The headers of column also contain target mRNA. Obviously, these new rearrangement will be more understandable by the readers.
Question (3): What does it mean “?” in Table 1? And what does it mean “Others” in Table 1?
Answer: Thanks for pointing out these inappropriate statements in old Table 1. We have deleted these questionable gene loci. Instead, only the definite genes associated with lupus pathogenesis are added in the new Table 1.
Question (4): What does it mean “↑or ↓” in Table 2? These should be described or defined. The authors should add “miRNA target gene, oxidative stress and epigenetic regulation” to the header of this column. I suggested the descriptions of miRNA and lncRNA in table 2 could be separated.
Answer: We have already separated old Table 2 into new Table 2 for miRs and new Table 3 for lncRNAs. We have also added footnotes to explain the meanings of these up- and down- regulated miRs and lncRNAs, and their respective targeted mRNAs. In addition, the oxidative stress-induced miRs are also asterisked and explained in the footnotes.
Question (5): Were lnc0597, lnc0949, lnc7074, lnc5150, lnc0597 and lnc0640 the new lncRNAs? I have never seen these descriptions. The references in Table 2 should be re-checked. For example, information of lnc7074, lnc5150, lnc0597 and lnc0640 were not described in reference 103.
Answer: We are extremely sorry these terrible typing errors. The correct names of these long non-coding RNAs would be linc0597, lnc0640, lnc5150 and lnc7074 in Table 3. We have also checked for the correct name for other lncRNAs in the same Table including linc0949, lnc7074, lnc5150, linc0597, linc-DC, and lnc0640 in the new version. The wrong cited reference has also been corrected to be Ref. 100 in the revised version.
Question (6): The descriptions of section 2.3 and introduction (page 3, lines 96-109) were similar.
Answer: We are sorry for the redundancy! We have already deleted the repeated statements in the revised version.
Question (7): Again, the name of ncRNA involved in exosome-mediated function should be described in detail. Which molecular/stimulator caused oxidative stress (OS) and which ncRNA was involving in OS-mediated function should be listed.
Answer: Thanks for these crucial questions. Similar to Question (1), we have added the names of ncRNAs involving in exosomal-mediated function and their target mRNAs or proteins in the newTable 2 and Table 3.
The molecules/stimulators induce oxidative stresses in SLE have been also added in Section 3.1. in the revised version:
“Lee et al [110] has extensively investigated the cause of excessive stress in patients with SLE. They reported that a number of anti-oxidant enzymes deficiency in the SLE leukocytes including copper/zinc superoxide dismutase (Cu/ZnSOD), catalase, glutathione peroxidase 4 (GPx-4), glutathione reductase (GR), and glutathione synthetase (GS). In addition, the mitochondrial biogenesis-related proteins such as mtDNA-encoded ND1 peptide (ND1), ND6, nuclear respiratory factor 1(NRF-1) and pyruvate dehydrogenase E1 component alpha subunit (PDHA1), and glycolytic enzymes including hexokinase II (HK-II), glucose 6-phosphatate isomerase (GPI), phosphofructokinase (PFK) and glyceraldehyde 3-phosphate dehydrogenase (GAPDH) are also reduced in SLE immune cells. These mitochondria functional abnormalities in the mitochondria may additionally increase oxidative stress, and cell apoptosis in patients with SLE in addition to defective bioenergetics”.
Regarding which ncRNA was involving in OS-mediated function? We have added some statements to explain in Section 4.1.:
“There has been no literature demonstrating direct evidence for specific oxidative-induced ncRNA in patients with SLE. The combination of Table 2 and Fig.5 let us speculate that miR-21, miR-29b, miR-146a and miR-126b are induced by excessive oxidative stress in SLE as asterisked in Table 2 and its footnote”.
Question (8): Figure 5 should be described in detail.
Answer: Thanks for this valuable and informative suggestion. We have already added two statements in Section 4.2. for explanation of Figure 5.
“Many authors have demonstrated that redox-dependent signaling is essential for host cellular decisions of differentiation, senescence or death to maintain homeostasis of the body [119-121]”.
The following paragraph is added for explaining no reference direct connection of miRs induced by oxidative stress in patients with SLE.
“However, despite the association of aberrant ncRNA expression with various pathological changes in SLE as listed in Table 2 and Table 3, there has been no literature demonstrating direct evidence for specific oxidative-induced ncRNA in patients with SLE. The combination of Table 2 and Fig.5 let us speculate that miR-21, miR-29b, miR-146a and miR-126b are induced by excessive oxidative stress in SLE as asterisked in Table 2 and its footnote”.
Minor comments:
Question (1): IFN- should be re-checked. There are several typing errors throughout the manuscript, which should be carefully reviewed by authors before submission. “miR-200C” should be “miR-200c” in Table 2. “miR-34C” should be “miR-34c” in Figure 5.
Answer: Thank you for pointing out these typing errors. We have already corrected these errors in the new version.
Reviewer 2 Report
Fundamentally, this is a well written review. Some major contributions, because important though lacking in the article, are recommended to be considered
1) Wang Y, Chen S, Chen S, Du J, Lin J, Qin H, Wang J, Liang J, Xu J. Long noncoding RNA expression profile and association with SLEDAI score in monocyte-derived dendritic cells from patients with systematic lupus erythematosus. Arthritis Res Ther. 2018 Jul 11;20(1):138. doi: 10.1186/s13075-018-1640-x.
2) Zhan Y, Guo Y, Lu Q. Aberrant Epigenetic Regulation in the Pathogenesis of Systemic Lupus Erythematosus and Its Implication in Precision Medicine. Cytogenet Genome Res. 2016;149(3):141-155.
Author Response
Answers to Reviewer (2):
Question (1): Fundamentally, this is a well written review. Some major contributions, because important though lacking in the article, are recommended to be considered: (1) Wang Y, Chen S, Chen S, Du J, Lin J, Qin H, Wang J, Liang J, Xu J. Long noncoding RNA expression profile and association with SLEDAI score in monocyte-derived dendritic cells from patients with systematic lupus erythematosus. Arthritis Res Ther 2018 Jul 11;20(1):138. doi: 10.1186/s13075-018-1640-x. (2) Zhan Y, Guo Y, Lu Q. Aberrant Epigenetic Regulation in the Pathogenesis of Systemic Lupus Erythematosus and Its Implication in Precision Medicine. Cytogenet Genome Res. 2016; 149 (3):141-155.
Answer: We are extremely sorry for missing these two important literatures in SLE pathogenetic and pathophysiologic studies. In the revised version, we have added literature-(1) Wang Y, et al. in Reference [105], Section 2.4. and new Table 3. A description in Section 2.4 emphasizes that:
“Wang et al. [105] found that increased lncRNA ENST00000604411.1 expression in macrophages/ dendritic cells that targets X inactive specific transcript (XIST) might involve X chromosome inactivation through protecting the active –X from ectopic silencing and induces lupus pathogenesis. Another lncRNA ENST 00000501122.2 (also known as NEAT1) overexpressed in in SLE-monocytes may activate IL-6, CXCL-10 and IL-6 expression. Furthermore, Wu et al. [100] reported that elevated expression of plasma GAS-5, linc 0640 and linc 5150 may activate MAPK signaling pathway. The five lncRNAs panel including GAS-5, linc7074, linc 0597, linc 0640 and linc 5150 in plasma could be applied as biosignatures in SLE”.
The literature-(2) Zhan et al. has been cited in Ref. [19] and in the section of “Introduction” for more completion.
Round 2
Reviewer 2 Report
The authors have met this Reviewer's recommendations